# Research on the relationship between Urban economic development level and urban spatial structure—A case study of two Chinese cities

Jun Zhang, Xiong He ⓘ *◉, Xiao-Die Yuan

School of Architecture and Urban Planning, Yunnan University, Yunnan, China

◉ These authors contributed equally to this work.
* ydxh@mail.ynu.edu.cn

**Data Availability Statement:** The data can be found within the paper and at DOI 10.17605/OSF. IO/SD59B.

## Abstract

The characteristics of urban spatial structure and the objective evaluation of the development level of urban economy have always been the concern of urban researchers, However, the spatial relationship between urban spatial structure and urban economic development level is often deliberately ignored. Through the point of interest (POI), the identification framework is constructed, the spatial structure of the city is identified and evaluated, and the Geographically Weighted Regression analysis is carried out with the distribution of unit GDP (Gross Domestic Product) in this study. The research shows that Kunming and Guiyang are polycentric spatial structures and Kunming's structure is more significant. Kunming's economic level is generally higher than Guiyang, but the unit area cannot be compared. The city center will promote the development of the central area in this city, and the more urban centers are distributed within the geographical and spatial range, the greater contribution would have to economic development. In addition, the results of this study will have a positive impact on urban planning and construction, and will also provide a new perspective for the study of cities and related disciplines.

## 1 Introduction

The process of urban diffusion and urban center formation in China is quite different from that in developed countries of Europe and the United States [1]. There are two main reasons: one is that Chinese government plays a leading role in urban planning and development. Another reason is that China's unique urban-rural isolation of the secondary land system is impeding spontaneous expansion of the cities [2]. Therefore, the spatial structure formation process of China's urban centers and its corresponding impact on cities have always been an important subject in academic fields such as urban planning and economic geography and so on [3,4]. At present, the study of urban spatial structure and urban polycentric has been widely spread, it covers from the identification [5] and assessment [6] of urban polycentric to the formation and evolution of urban spatial structure, etc. [7,8].

 

**Funding:** The author(s) received no specific funding for this work.

**Competing interests:** The authors have declared that no competing interests exist.

There are a variety of methods to identify and evaluate urban polycentric spatial structures for different city size. In this paper, the author identifies the spatial structure of urban polycentric from the perspective of human activities with the help of the mature method framework and point of interest data [9]. And it also uses two typical cities in western China as the points to answer the positive impact of urban spatial structure features on urban development.

## 2 Literature review

### 2.1 Identification data and methods of urban spatial structure

Those traditional urban research data, including statistical yearbook data and census data, show great inadaptability in today's urban studies [10], they mainly behave in the difficulty of obtaining data and the long renewal cycle [11,12]. In recent years, the use of urban big data has provided a new research perspective and paradigm on urban research [13], the current mature urban data include mobile phone signaling data [14], network review data [15], rail traffic card data [16], thermodynamic chart data [17], and so on. Scholars from various countries also have used these data to explore urban space and summarized many new theoretical and methodological models [18,19]. Based on China's actual national conditions, the urban data are in a less accessible position and most of the data are confidential, which have puzzled many researchers, so Chinese young scholars begin to explore a new geographical open source data POI (Point of Interest) for urban space exploration as an alternative [20,21]. First of all, POI data is the abstract expression of real geographical entities in virtual space, it is the one not only with spatial attribute information of geographical entities but also has large amount and easy to obtain. POI has become one of the important data in urban research [22]. At present, scholars find that POI performs very well in exploring the formation of urban spatial structure [23] and the identification of the urban functional area [24].

There are many methods of urban spatial recognition, including morphology [25], geography [26], economics [27], and many other disciplines, but different methods are applicable to different data types, and this study emphasizes on the urban population and the activities they have.

In 1991, Giuliano pioneered the idea of identifying urban centers through the employment density of the regional population. This method primarily focused on setting the minimum employment figure and employment density of urban centers, then identified urban centers by the differences between validated thresholds and research results [28], which was widely used in the subsequent researches [29]. In 2012, Agarwal proposed a city center recognition method based on the distribution of continuous density ladder values, it set the highest density as the primary center and local peaks as the sub-center. The method of identifying that based on density is quite subjective [30],because this needs the researchers have a more detailed understanding on the study area to ensure the accuracy of local peak selection, so this approach was inefficient and inaccurate in applying in more cities [31,32]. In recent years, the identification of urban centers and their structures through thermodynamic chart and gridded population distribution has become applicable to most cities and regions [33,34] but if we combine it with the current methods of urban center identification, there are still great subjectivity and uncertainty in the exploration of cities [35].

### 2.2 Evaluation of urban development level

How to assess the level of urban development quickly and accurately is the thing that needed to be understood correctly by government policy makers and urban construction [36]. Urbanization assessment can have different understanding from different disciplines, in economics, the most obvious change in the level of urban development is the change of regional GDP,

which can measure the economic level of a region and compare the development of different cities [37]. In sociology aspect, the most significant sign of the level of urban development is the change of floating population in the region, the greater the change of the floating population over a certain period of time is, the higher level of urban development will be [38]. In urban geography, the higher level of urban development is, the greater the construction area of the city will have, so the level of urban development can be reflected by changes of the impermeable ground [39]. The advantages of urban development level methods reflected by different disciplines are different, and this study hopes to show the level of urban development within the fine range scale, so the paper will combine the economic level with the perspective of urban geography [40], and express the regional level of urban development objectively through the GDP per unit area [41].

### 2.3 Geospatial association analysis

The trend of urban spatial structure relates to the correlation of urban geographic space, which is a challenge, and the problem is that static or aggregated measurements cannot reveal the clustering effects of geographic space [42]. Therefore, a method is needed to analyze the relationships of change caused by geographical factors [43]. At present, the popular methods in geospatial to reveal is the spatial statistical analysis, such as spatial self-correlation analysis [44], global statistical analysis [45], local statistical analysis [46], etc., but these methods tend to express the correlation of geospatial location and geospatial trend assessment, they only consider themselves in the geographical spatial distribution [47]. In this study, it is hoped that the spatial changes under a certain scale and the related driving factors can be explored and used to predict urban development. So, through comparing the literature [48,49], the geo-weighted regression analysis is used to describe it, since this analysis takes into account the local effects of spatial objects, it is more accurate than the rest of the space analyses [50].

## 3 Research areas and data

### 3.1 Study area selection

As the capital city of Yunnan Province, the central city of the Central Yunnan urban agglomeration, and one of the most important central cities in western China approved by the State Council, Kunming has a resident population of 4.230 million and and the third highest GDP growth rate in the country, reaching 75.565 billion USD by 2018. Like Kunming, Guiyang, the capital of Guizhou province, is also a central city in western China. By 2018, Guiyang has a resident population of 3.136 million and a GDP of 53.672 billion USD, ranking second in the country in terms of growth rate.Western cities of China are located in the interior and their developments are slow, since March 2015, the National Development and Reform Commission and other departments jointly released the "Vision and Actions on Jointly Building Silk Road Economic Belt and 21st-Century", referred to as "the Belt and Road", it brought the historical development opportunities for western cities in, Kunming and Guiyang's urban developments have also ushered in new development goals and requirements.

The official release of the "Immediate Plan (2016–2020) and the Land and Space Plan (2018–2035) of 2017 in Kunming and the Guiyang City Master Plan 2011–2020 (2017 Revision) proposed to make the capital of the province to be the political, economic, cultural, scientific & educational and tourism center, and became the important and sustainable cities in western China (Fig 1).

Regarding the study of urban spatial structure and the level of urban economic development, Shanghai and Guangzhou in China have always been the research hotspots, while the research on western China is relatively scarce, especially in the area of the Yungui Plateau.

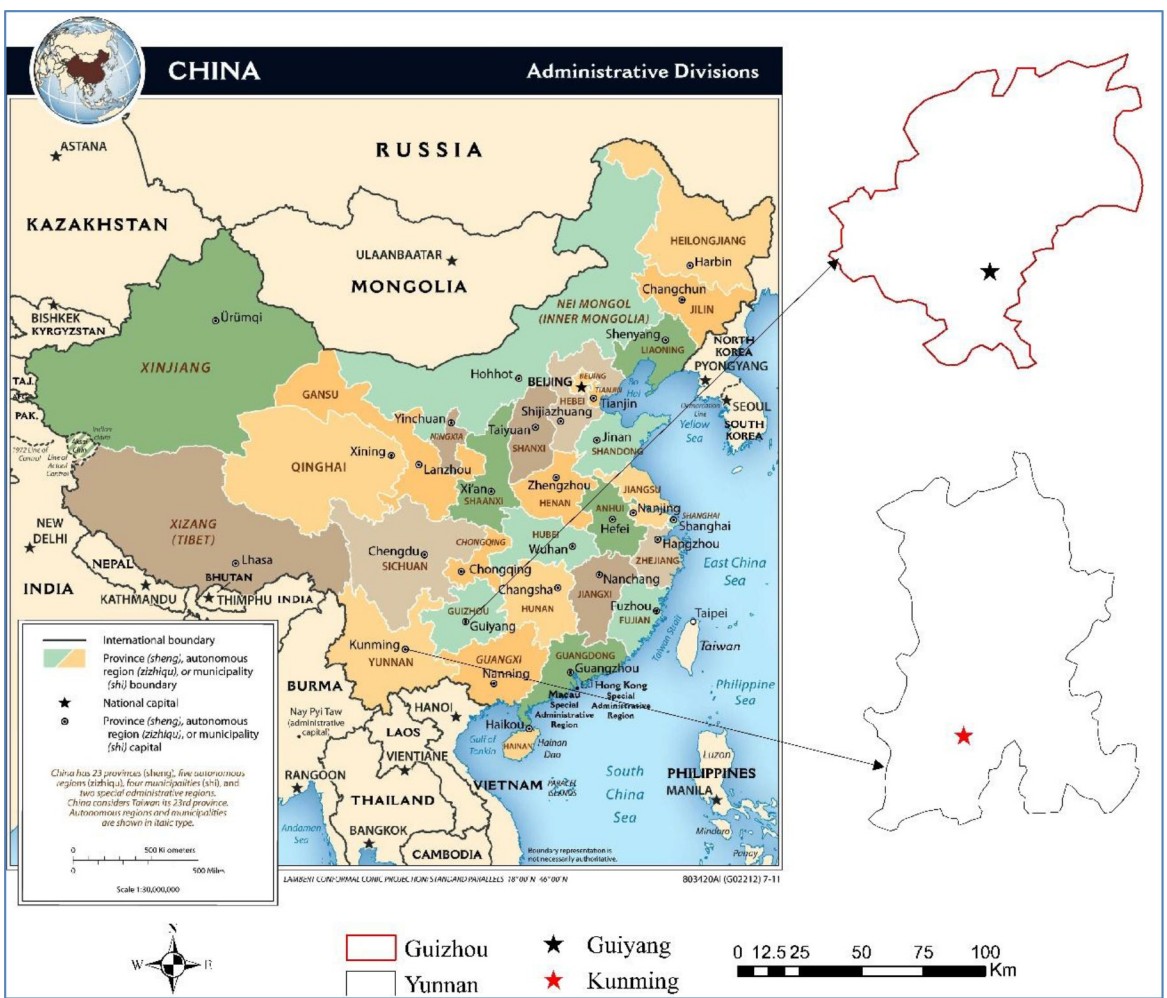

**Fig 1. Study area.** (Image Source: https://www.cia.gov/library/publications/resources/cia-maps-publications/China.html).

With the implementation of the "One Belt and One Road" policy, the urban built-up areas and the economic level of Kunming and Guiyang have achieved rapid development in just a few years, which calls for more case analysis and research. In addition, the leaders and planners of Kunming and Guiyang all agree that although their cities are developing very well in all respects, they are still lack of relatively objective reasons to prove it. Therefore, the double discussion of urban spatial structure and urban economic development level of the two cities carried by this study will also be conducive to the answer of this practical question.

## 3.2 Data acquisition and processing

The data are the points of interest data of Kunming and Guiyang in 2016 and the GDP grid data from 2016 to 2018, respectively. POIs are provided by Amap Online (www.amap.com). In this study, the POI data is obtained through the open API interface provided by Amap, and the data is verified and geocoded. This study believes that POI contains all the urban functions of the research area. The reason is that POI has the spatial attributes and location information of the urban object entities. Depending on the different functions offered by the city, road ancillary facilities, public facilities, transportation facilities, access facilities, government

**Table 1. POI category and quantity of Kunming and Guiyang in 2016.**

| Category | Subcategory | Guiyang | Kunming |
|---|---|---|---|
| Public | Road ancillary facilities | 4958 | 7884 |
| | Public facilities | 21963 | 30981 |
| | Transportation facilities | 9881 | 16642 |
| | Access facilities | 3269 | 4331 |
| | Government agencies and Social groups, | 897 | 788 |
| | Science and education Cultural services | 1907 | 2001 |
| Living | Catering services | 100932 | 156831 |
| | Shopping services | 23914 | 30002 |
| | Health care services | 1003 | 1412 |
| Leisure | Resort sanatoriums | 2973 | 4089 |
| | Golf | 34 | 63 |
| | Entertainment venues | 10034 | 19387 |
| | Sports venues | 875 | 1067 |
| Residential | Community | 5903 | 7841 |
| | Villa areas | 580 | 890 |
| | Dormitory buildings | 1820 | 2045 |
| Business | Companies | 7001 | 8901 |
| | Enterprises | 6603 | 7102 |
| | Bank | 781 | 561 |

agencies and social groups, science and education cultural services, catering services, shopping services, health care services, resort sanatoriums, golf-related, entertainment venues, sports venues, community, villa areas, dormitory buildings, companies, enterprises, banks, a total of 19 types of POI are divided into 5 city functions, including public functions, living functions, leisure functions, residential functions and business functions. The specific categories and quantities are in Table 1.

GDP (gross domestic product) refers to the market value of all final products and services produced by all resident units in a country or region over a certain period of time, and GDP is the basic indicator of macroeconomic development. The GDP km grid data are all derived from the Geographical Information Monitoring Cloud Platform (http://www.dsac.cn/DataProduct). The GDP grid data break the traditional expression of distribution with administrative division as the unit, they use statistical GDP data (including the first, second and third industrial structures), and land use type data to build a spatial relationship model, then with the support of elevation, geomorphology, etc. they generate 1km x1km GDP grid data to reflect the distribution of GDP within the administrative unit.

## 4 Research methods

This study has two main objectives: first, is to identify the urban spatial structure of Kunming and Guiyang, second, is to explore the spatial connection between spatial structure and the development level of urban economy.

For the first goal, the urban spatial structure is identified from the peak of the point density of POI (the sum of the number of POI in a defined area). Point density within the city is first calculated, hoping to generate continuous surface of the density in the territory, then it is divided into 10 categories with Natural Breaks (Jenks) (Natural Breaks can maximize the difference between each category), On each surface, it is determined whether the area is an urban center according to the difference in density area. When the density difference reaches 20%,

the density change is irreversible. In other words, the area where the density surface difference is greater than 20% is determined as the city center. For the second target, although the spatial panel model and the minimum binary model used in the previous study could deal with the relationship between different regional spaces and their associated influences, we first need to assume that the relationship would not change with the spatial position, so it is necessary to consider spatial heterogeneity [48]. Geographically weighted regression (GWR) model embeds spatial position regression parameters and reflects the spatially unstable nature of the parameters in different regions, thus it improves the relationship that variables can vary with spatial location [50]. The model is more realistic, and the functional relationship of geographically weighted regression can be expressed as:

$$y_{i=\beta_0(U_i,v_i)+\sum_{k=1}^{P}\beta_k(U_i,v_i)x_{ik}+\varepsilon_i} \quad i=1,2,3......n \tag{Eq 1}$$

where $y_i$ represents the dependent variable; $x_{ik}$ denotes the explanatory variables of an n × n matrix; $\beta_k$ indicates the parameter vector to be estimated; $(U_i, v_i)$ is a function of location; $k$ stands for the number of parameters to be estimated; $i$ represents $i$th region; and $\varepsilon_i$ is a random error term of $i$th area.

The theoretical model of this method can be explained as:

## 5 Research results

### 5.1 Identification of POI-based urban spatial structure

The results show that the urban space of Kunming and Guiyang in 2016 represent an obvious polycentric structure, and the spatial structure of the two cities is recognized based on the morphological dimension of 2016.

**5.1.1 Number of urban centers.** In the point density map, there is only one urban center in the large-scale urban area, and the number of urban centers in the two cities of that year is determined by the region and the calculation of density ratio. As can be seen from Fig 3, the number of urban centers in Guiyang is 5, while Kunming has 8, so from the number of urban centers, Kunming has more obvious urban polycentric structure compared to Guiyang.

**5.1.2 Urban center form.** Urban center form can refer to the land use scale and land use intensity, Land use is specifically expressed by FAR(Floor Area Ratio), which is the ratio of the total area to the average area. [51], by contrast, ratio of the city center's total area to POI density area can be used to replace the ratio of total area to average area. Although the density area of POI is not equivalent to FAR, since the change of POI in urban space does not affect the change of FAR directly, POI can express the full function of the city, so the regional POI density area is more important and immediate to the morphological expression of urban centers. Since the POI density distribution in the entire urban space is diverse, the urban centers identified in Fig 2 are selected to calculate their regional POI density area ratio.

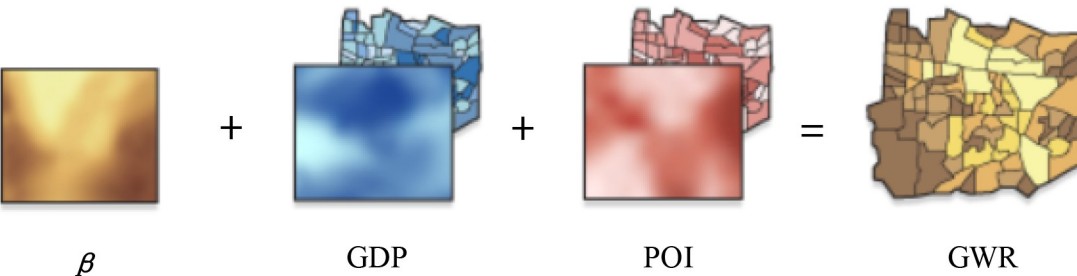

β GDP POI GWR

**Fig 2. Theoretical model of GWR.**

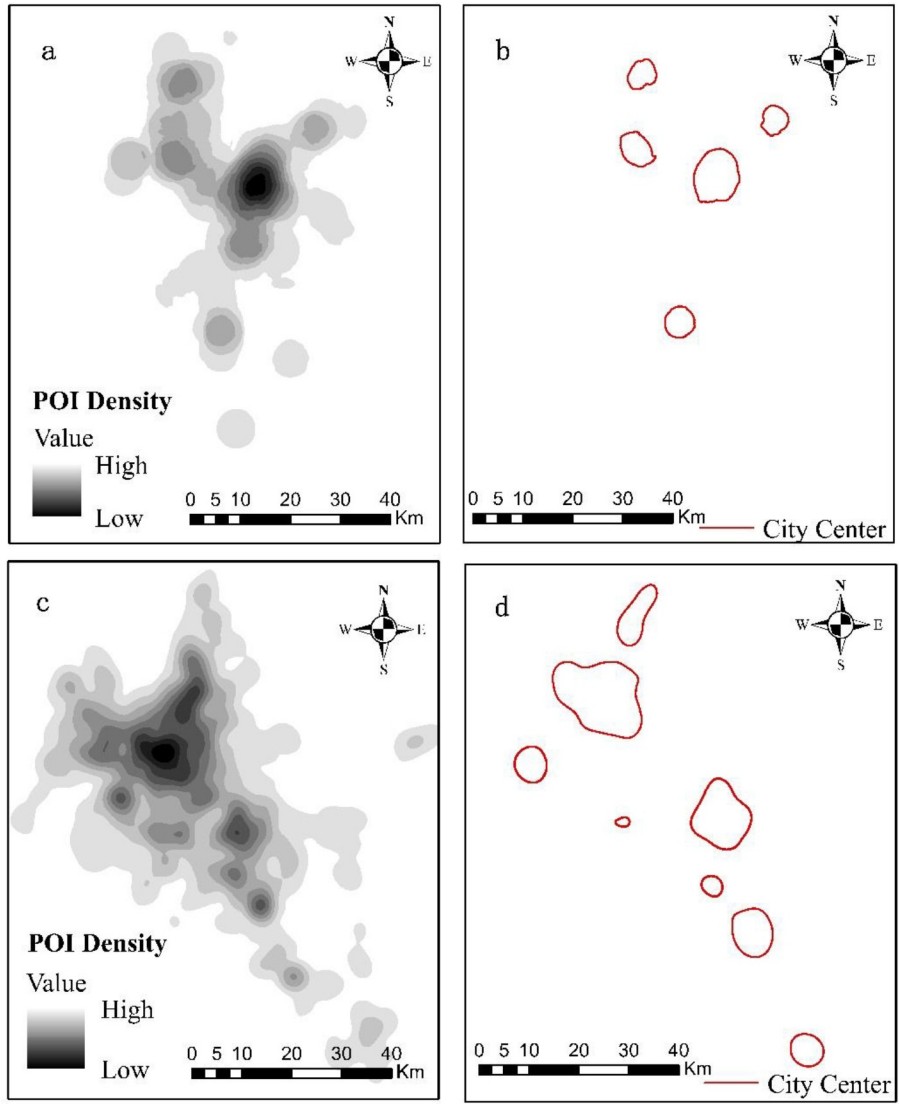

**Fig 3.** Identification of polycentric cities (b, d) in Guiyang and Kunming (a, c) in 2016.

Urban center size: Compared to the size of the two urban centers, Guiyang has 5 urban centers, they are relatively mature, the vertical distance is about 50 km, the largest area of a single urban center accounts for 28.63%, the urban center area accounts for 19.66% of the total area. There are 8 urban centers in Kunming with a vertical distance of more than 90 km, six urban centers are relatively mature, and the rest of them are in the embryonic stage, the largest area of a single urban center accounts for 34.18% and the urban center area accounts for 36.52% of the total area. From the scale point of view, Kunming's urban polycentric space structure is larger.

Urban Center Intensity: The POI density in urban centers is significantly higher than in other regions, and by calculating the surface difference ratio of urban center density, we can know that the highest ratio of the five urban centers in Guiyang is 2.122, and the lowest is only 1.391. The ratio of the eight urban centers in Kunming is as high as 5.639, and there are 3 urban centers whose ratio is over 2.122. The denseness of POI reflects the rapid development

of urban centers, and the form of it represents that the urban center's POI agglomeration is faster than the surrounding area, this intensity's difference causes the difference in the surrounding "attraction". From the intensity, we can see that Kunming's urban polycentric spatial structure strength is higher.

## 5.2 Evaluation of urban development level based on GDP

Total GDP: According to the 2016, 2017 and 2018 China Domestic GDP Reports, Guiyang's GDP is respectively 45.11 billion USD, 51.97 billion USD and 54.264 billion USD, while Kunming's GDP is 61.435 billion USD, 69.395 billion USD and 74.384 billion USD. In terms of overall GDP, Kunming's economic output is greater than Guiyang's in these three years, and the annual increase in GDP is also greater than Guiyang, but from China's GDP report, Kunming and Guiyang GDP growth rate is declined, that maybe because of the affection by the "The new normal of China's economy", which is manifested in China 's transition from traditional GDP growth to sustainable GDP growth, resulting in a slowdown in GDP growth. Based on the total GDP and annual increase of the two cities, Kunming's urban development level is higher than Guiyang's.

Unit GDP: Unit GDP shows the distribution of GDP per square kilometer (Fig 3), as can be seen from the figure that, Guiyang's highest unit GDP is relatively 14.062 USD, 16,703 USD, 21,047 USDfrom 2016 to 2018, while Kunming's is 15,689USD, 19,520USD and 22,285 USD. Guiyang's GDP-generating area accounts for 50.33%, 56.21% and 61.03% of the total administrative area, while Kunming's accounts for 33.19%, 35.44% and 39.86% of the total administrative area respectively. Guiyang's high unit GDP accounts for 17.33%, 16.29%, 13.75% of the total area, while Kunming's accounts for 22.64%, 22.99% and 24.10% of the total area, respectively. In terms of unit GDP, Kunming's three-year maximum unit GDP and high-value ratio are both higher than Guiyang's, but the producing GDP area is smaller than Guiyang. Based on the distribution of unit GDP, Kunming's economic region is more concentrated than Guiyang's.

## 5.3 Geographically weighted regression -based analysis results

As shown in Fig 4, the regression coefficient of the adjacent unit grid has maintained a similar level, so the development of economy of the city and the spatial structure of the city have a high spatial dependence. As can be seen from Fig 4, there is an obvious positive correlation between the urban spatial structure and the level of urban economic development, which shows that the urban spatial structure has a one-way guiding effect on the level of urban economic development. In the process of urban development, with the continuous agglomeration and diffusion of urban elements, one city center after another is formed, and the urban spatial structure changes along with it. The attraction of urban centers to people and their corresponding activities is far greater than that of other regions in the city. Therefore, with the increase of people and economic activities, consumption is bound to be driven, which will undoubtedly lead to the economic development level of urban centers surpassing that of other regions.

From the distribution region of the regression coefficient, the distribution of high value of the regression coefficient in Guiyang increased from 2016 to 2018, while the distribution of low value decreased, but the overall distribution of the regression coefficient showed a trend of gradual decrease. From the point of view of the concentration of high and low values of the regression coefficient, the distribution of high values of the regression coefficient in 2016 shows as a single central concentration, the regression coefficient begins to spread in 2017, and the distribution of high values in 2018 shows as multi-central concentration. From the

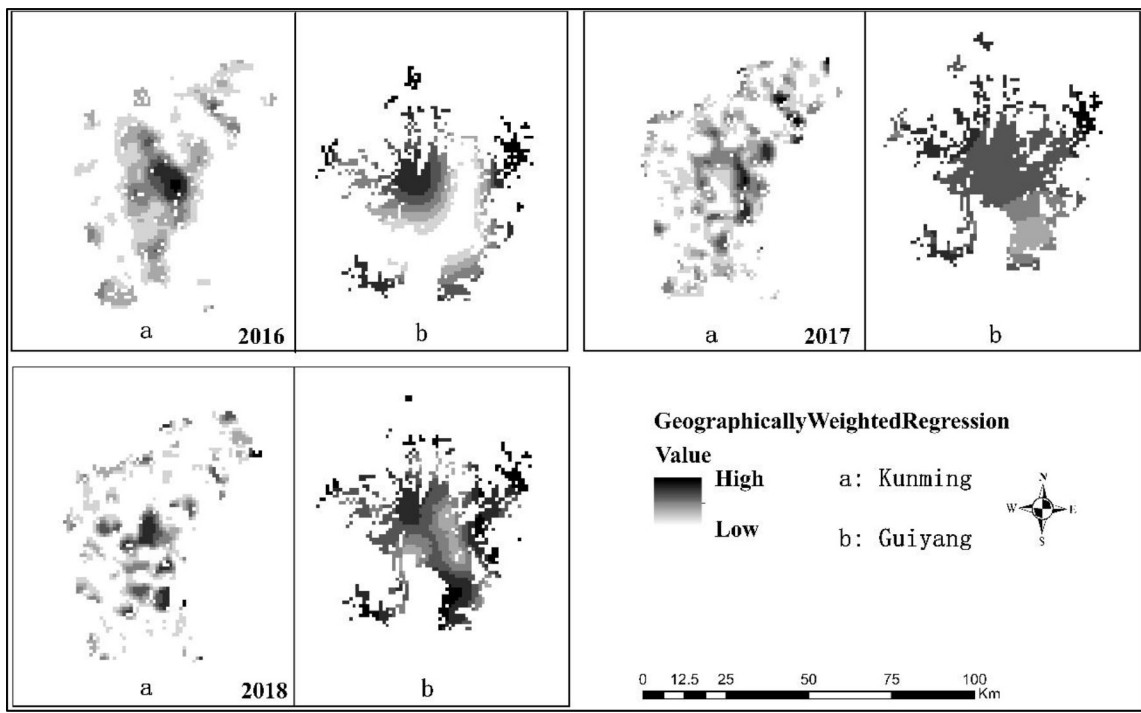

**Fig 4. Results of GWR analysis in Guiyang and Kunming from 2016 to 2018.**

distribution of GWR in Guiyang from 2016 to 2018, the influence of urban spatial structure on urban economic level is positively correlated.

Similar to the coefficient distribution in Guiyang, from 2016 to 2018, the distribution of high value GWR regression coefficient in Kunming also increased gradually, while the distribution of low value gradually decreased, but the overall distribution of high and low value did not show an overall trend of decrease. As shown in Fig 5, the urban spatial structure of Kunming is more complex than that of Guiyang, and there are more urban centers. As a result, the linkage effect between the urban center and surrounding areas is enhanced. At the same time, the development of the urban economy is no longer limited to the city center, but the city center drives the surrounding areas to achieve common development. The high-value agglomeration of regression coefficients from single-region agglomeration to multi-region agglomeration also illustrates this problem. From an objective comparison point of view, Kunming's urban spatial structure is more complicated than that of Guiyang, resulting in a higher level of urban development in Kunming.

The increase of GWR coefficient in kunming and guiyang from 2016 to 2018 indicates that the spatial correlation between urban spatial structure and urban development level is strengthened. The spatial structure of the two cities of Guiyang and Kunming is multi-center, and the change of the regression coefficient is also concentrated in the center of the city. Moreover, compared with Guiyang, Kunming has more urban centers, more complex spatial structure and better urban development level. Therefore, it can be concluded that the multi-center urban spatial structure is essentially affecting the development of the city.

## 6 Discussion

This study uses POI big data to identify his study uses POI big data to identify the urban spatial structure of Guiyang and Kunming, two typical cities in Western China, and to explore the

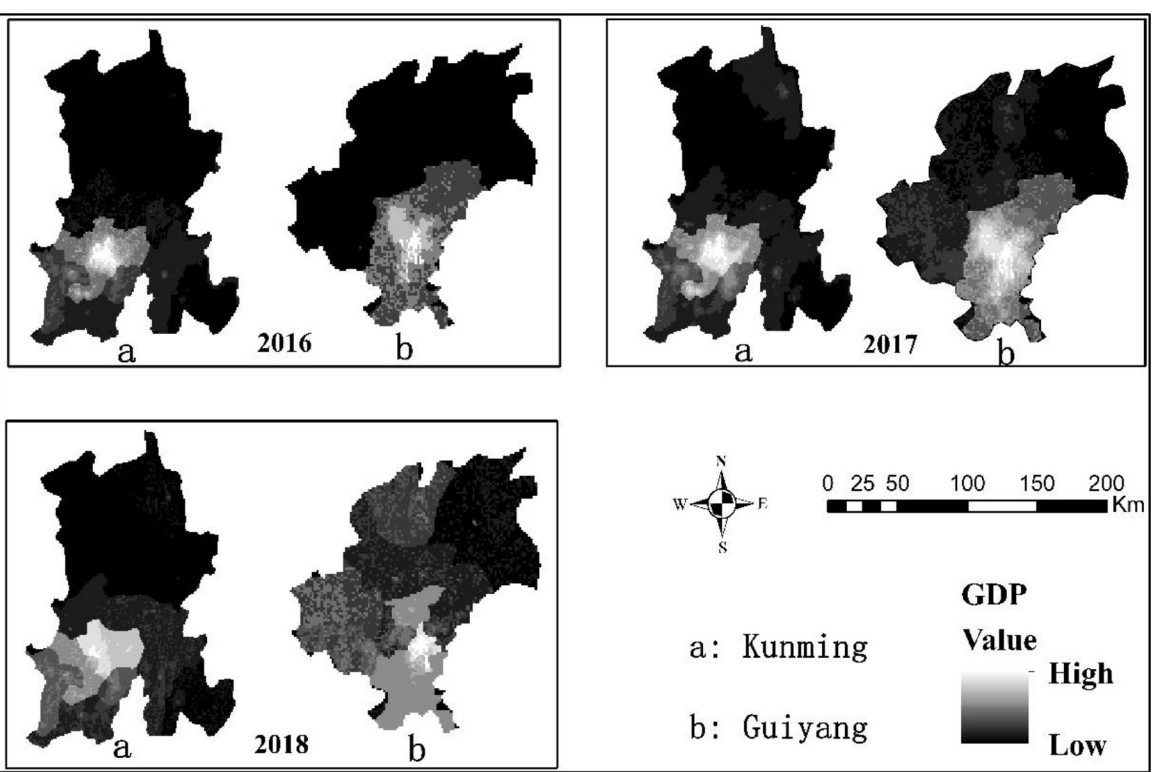

**Fig 5. Guiyang and Kunming unit GDP from 2016 to 2018.**

impact of urban spatial structure on the level of urban economic development by connecting with economic data. The research holds that firstly the urban spatial structure identified by POI big data makes up for the inability and identify and judges accurately due to the subjective nature of the previous data, and also avoids some errors in the previous research methods, it mainly focuses on the regional threshold and the scale effect under the spatial scale set by subjective arbitrariness, which makes the results of the recognition of urban spatial structure more credible. Secondly, instead of using administrative boundaries as a research area in the study of Chinese cities, it is replaced with the geographical and practical space of cities, which makes the results of the analysis more objective and, to some extent, explains the difference between Chinese cities and cities in Europe and the United States. Finally, the development level of the city is expressed by the unit GDP, and the spatial relation between it and the urban spatial structure is analyzed, which provides a new research perspective for the subsequent urban research.

Obviously, there are still some shortcomings in this study, and have room for continued improvement. First of all, although the study uses big data for research, the identification of urban spatial structure method is still replaceable, for example, through remote sensing image classification to identify urban structure [52]. Second, POI itself is only a geographical coordinate, the urban entities in place in the actual geographical space in the city is not homogeneous, and in the first part of the identification of urban spatial structure, there is no distinction between treatment, nor does it take into account the need for weighted calculation. In addition, the level of urban development as indicated by GDP is objective but not comprehensive, for example, Kunming's tourist floating population is much larger than Guiyang, but the unit GDP does not take into account the impact of population movement on urban

development. In addition, contrary to the multicollinearity analysis of urban spatial structure and urban economic development level affected by multiple factors, this study only studied the correlation between urban spatial structure and urban economic development level. Finally, the time span of this study is only three years, which is relatively shorter for the "development" of a city's spatial structure. Due to the fact that the POI data itself has the disadvantage of short data acquisition time, multi period remote sensing data fusion and comparative analysis will be added in the next study. The focus of this study is to identify the relation between urban spatial structure and urban development, so these deficiencies will be explored continuously in the next study.

Exploring urban spatial structure with POI big data is a new paradigm of urban research, and exploring its objective impact on urban development through the difference of urban spatial structure is a new urban research perspective, it will produce great changes in thought and method of exploring urban planning, urban construction, and urban economy. Meanwhile, new innovations mean new challenges, this study is only a preliminary one, and we still need more researches to deepen and complement the ongoing theory and methods.

## 7 Conclusion

Based on the POI data provided by The AMap, this study identifies the urban spatial structure which is of great significance to urban planning and urban construction, and makes morphological analysis of this spatial structure. Through the analysis of economic data, the development status of Kunming and Guiyang in China is judged, and the comparison of the two cities is made by the geographical weighted regression analysis. The following conclusions are drawn:

First, Kunming and Guiyang have shown obvious polycentric spatial structure since 2016, and through morphological analysis, Kunming's polycentric city spatial structure is higher in intensity or scale than Guiyang, so Kunming's polycentric spatial structure is more significant than Guiyang.

Second, through the analysis of economic data, Kunming's urban economic development level is higher than Guiyang, unit GDP is also higher, but because of different regions of industry and labor services, the level of development in the inner region of the city does not have horizontal comparability.

Third, although Kunming's urban spatial structure and urban economic development level are higher than Guiyang, it does not mean that the two have a clear two-way relation. Instead, the analysis of geographically weighted regression shows that the polycentric urban spatial structure will promote the economic development in the unit area, and the more obvious the polycentric structure of urban space is, the more obvious the effect of promoting will be.

By summarizing the results of this study, we can get a general evaluation of China's urban spatial structure and urban development, and the results will objectively guide the planning and construction of cities to some extent.

## Author Contributions

**Data curation:** Xiong He.

**Formal analysis:** Jun Zhang.

**Methodology:** Xiong He.

**Software:** Jun Zhang.

**Visualization:** Xiao-Die Yuan.

Writing – **original draft:** Jun Zhang.

Writing – **review & editing:** Xiong He.

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
