## [Decision Letter · Decision Letter 0]

20 Apr 2020

PONE-D-20-08232

Research on the Relationship between Urban Economic Development Level and Urban Spatial Structure—A Case Study of Two Chinese Cities

PLOS ONE

Dear Dr. He,

Thank you for submitting your manuscript to PLOS ONE. After careful consideration, we feel that it has merit but does not fully meet PLOS ONE’s publication criteria as it currently stands. Therefore, we invite you to submit a revised version of the manuscript that addresses the points raised during the review process.

We would appreciate receiving your revised manuscript by Jun 04 2020 11:59PM. To enhance the reproducibility of your results, we recommend that if applicable you deposit your laboratory protocols in protocols.io, where a protocol can be assigned its own identifier (DOI) such that it can be cited independently in the future. For instructions see: http://journals.plos.org/plosone/s/submission-guidelines#loc-laboratory-protocols

We look forward to receiving your revised manuscript.

Kind regards,

Bing Xue, Ph.D.

Academic Editor

PLOS ONE

Journal Requirements:

2. We note that the Figure in your submission contain [map/satellite] images which may be copyrighted. All PLOS content is published under the Creative Commons Attribution License (CC BY 4.0), which means that the manuscript, images, and Supporting Information files will be freely available online, and any third party is permitted to access, download, copy, distribute, and use these materials in any way, even commercially, with proper attribution. For these reasons, we cannot publish previously copyrighted maps or satellite images created using proprietary data, such as Google software (Google Maps, Street View, and Earth). For more information, see our copyright guidelines: http://journals.plos.org/plosone/s/licenses-and-copyright.

a)    You may seek permission from the original copyright holder of the Figures to publish the content specifically under the CC BY 4.0 license.  

3. In your Data Availability statement, it is unclear why 'No - some restrictions will apply' please clarify the nature of these restrictions. PLOS defines a study's minimal data set as the underlying data used to reach the conclusions drawn in the manuscript and any additional data required to replicate the reported study findings in their entirety. All PLOS journals require that the minimal data set be made fully available. For more information about our data policy, please see http://journals.plos.org/plosone/s/data-availability.

Reviewers' comments:

Reviewer's Responses to Questions

**Comments to the Author**

1. Is the manuscript technically sound, and do the data support the conclusions?

Reviewer #1: Yes

Reviewer #2: Partly

Reviewer #3: Yes

Reviewer #4: No

2. Has the statistical analysis been performed appropriately and rigorously? 

Reviewer #1: Yes

Reviewer #2: Yes

Reviewer #3: N/A

Reviewer #4: No

3. Have the authors made all data underlying the findings in their manuscript fully available?

Reviewer #1: No

Reviewer #2: No

Reviewer #3: Yes

Reviewer #4: No

4. Is the manuscript presented in an intelligible fashion and written in standard English?

Reviewer #1: Yes

Reviewer #2: Yes

Reviewer #3: Yes

Reviewer #4: No

5. Review Comments to the Author

Reviewer #1: The subject is essential for urban designers and planners. I believe the authors did a good research paper. They highlighted the importance of considering the GDP to understand the urban structure in addition to providing a method to compare different centers in a city. I enjoy the ready. The paper is well organized with good English. I have several comments as follow:

1. Two acronyms, GDP and GWR, should be defined once appeared in the text.

2. The Keywords did not indicate the most relevant keywords for the research, such as “point density beak and FAR” that used for data analysis.

3. Under the title, “study area selection,” I have two comments:

• The authors indicated that the residential population of Kunming was 499.02 million people, and of Guiyang was 48.819 million people. About the following link: https://populationstat.com/china/kunming, the population of Kunming in 2020 is 4,4443,000. I believe there is a mistake in the number of population of both cities.

• In the same paragraph, the author mentioned the GDP in Yuan, the Chinese currency. Since most international readers are not familiar with its value, I suggest adding the equivalent of the GDP value in USD. To have a good idea about the importance of the GDP of both cities to China, it is crucial to indicate the GDP of China, which is 12.24 trillion USD in 2017, according to the flowing link: https://www.google.com/search?q=gdp+of+china&rlz=1C1CHBD_enQA842QA842&oq=GDP&aqs=chrome.1.69i57j0l7.12365j0j7&sourceid=chrome&ie=UTF-8

4. In the section of the “Research Methods section,” I have four questions:

• In the second paragraph, the authors used the term “natural interruption.” Did they mean natural separators like a river, mountain, and valley? Since we are dealing with the city as a human product, what about the artificial interrupters, like railways, main artery roads, parks, etc.?

• Also, in the same paragraph, the author referred to 20 % as a determinate factor for the city center, why 20 %? I suggest adding justification.

• In Paragraph 3 (at the end of the same page) line 2, the author stated that the “previous study could deal…”, which previous study the authors referring to?

• Eq. 1 is the basic equation used by the authors for data analysis. I suggest showing an example of the application.

5. In the section of the “research results,” paragraph 4, line 2, the author referred to the relative dimension of the urban center as vertical distance. Why vertical distance and not horizontal? Here the author talking about the physical dimension and not a time-space perspective.

6. On the next page of the same section, the author refers to the phrase: “the new normal of china’s economy.” I suggest writing a statement explaining what the “new normal of china’s economy is”.

Best luck

Reviewer #2: 1.The urban spatial structure is affected by many factors. It is not sufficient to choose the economic development level as the influencing factor. And the selected typical cities Guiyang and Kunming are not representative.

2.For the urban economic development level and urban spatial structure, the three-year time dimension is too short to reflect its real change.

3.Amap is an online map application. Its data acquisition methods and data analysis methods are not described in sufficient detail. Thus, it is impossible for readers to repeat the research according to the article.

4.There are many mistakes and omissions in the article, such as the population "499.02 million people".

5.The illustrations in the article are not standardized, and the font size is not uniform. Some Chinese characters appear in some parts of the illustrations.

Reviewer #3: Manuscript Number: PONE-D-20-08232

Article Type: Research Article

Full Title: Research on the Relationship between Urban Economic Development Level and Urban Spatial Structure—A Case Study of Two Chinese Cities

This is nice work carried out by authors on the spatial relationship between urban spatial structure and urban economic development level. I have few comments on it as below:

1. On Fig. 1, it would be better to connect Guiyang and Kunming from China’s map to enlarged individual map of city by ‘arrow’

2. The data are Kunming and Guiyang's POI data of 2016 and GDP grid data of 2016-2018: Can you provide citation or website associated with this sentence?

Reviewer #4: The relationship between urban spatial structure and the economy is very complex. Generally, the economy development depends on the location, the population, the industrial level., and the urban spatial structure. This paper presents a case study to examine the association between them. It has several shortcomings:

1. The multicollinearity of multivariate regression analysis are not addressed.

2. The results is not clear, especially those in Figure 4. Which facror does it represent for?

6. PLOS authors have the option to publish the peer review history of their article (what does this mean?). If published, this will include your full peer review and any attached files.

Reviewer #1: Yes: M Salim Ferwati

Reviewer #2: Yes: He Ying

Reviewer #3: No

Reviewer #4: No

---

## [Author Response · Author response to Decision Letter 0]

22 May 2020

I have edited the full text of the editor's opinion directly. For the detailed part, please see the full text.，and the new map comes from: Maps at the CIA (public domain): https://www.cia.gov/library/publications/the-world-factbook/index.html and https://www.cia.gov/library/publications/cia-maps-publications/index.html. Recently changed：The map in Figure 3 4 5 has been deleted, and now Figure 3 4 5 are all images automatically generated on the GIS, and have nothing to do with the map. And the data in Figure 3 4 5 has been uploaded to the OSF(10.17605/OSF.IO/SD59B), you can check the connection mentioned above for details.Now the maps 3 4 5 have no map images, no base map!

Reviewer #1:

1、Two acronyms, GDP and GWR, should be defined once appeared in the text.

Respond: In the paper, we defined GDP and GWR where they first appeared, with GDP first defined in Abstract and GWR first defined in Chapter 4.

2、The Keywords did not indicate the most relevant keywords for the research, such as “point density beak and FAR” that used for data analysis.

Respond: In this paper, we have supplemented the definition of some keywords including point density beak and FAR, in which point density beak stands for the sum of the number of POI within the limited Area, and FAR for Floor Area Ratio.

3、 Under the title, “study area selection,” I have two comments:

The authors indicated that the residential population of Kunming was 499.02 million people, and of Guiyang was 48.819 million people. About the following link: https://populationstat.com/china/kunming, the population of Kunming in 2020 is 4,4443,000. I believe there is a mistake in the number of population of both cities.

In the same paragraph, the author mentioned the GDP in Yuan, the Chinese currency. Since most international readers are not familiar with its value, I suggest adding the equivalent of the GDP value in USD. To have a good idea about the importance of the GDP of both cities to China, it is crucial to indicate the GDP of China, which is 12.24 trillion USD in 2017, according to the flowing link: https://www.google.com/search?q=gdp+of+china&rlz=1C1CHBD_enQA842QA842&oq=GDP&aqs=chrome.1.69i57j0l7.12365j0j7&sourceid=chrome&ie=UTF-8

Respond: For the first population problem, it is true that the data we obtained is not the same as the data you provided. The reason is that the time point of this study is 2019, so we used the population data of Kunming and Guiyang in 2018, which are 4.23 million and 3.136 million respectively. In order to facilitate your review, we have unified the population data into the data of the website you provided. As for the second question: GDP, we have converted all the GDP data obtained in 2018 into US dollars, which are US $75.565 billion and US $53.672 billion respectively.

4、In the section of the “Research Methods section,” I have four questions:

In the second paragraph, the authors used the term “natural interruption.” Did they mean natural separators like a river, mountain, and valley? Since we are dealing with the city as a human product, what about the artificial interrupters, like railways, main artery roads, parks, etc.?

Also, in the same paragraph, the author referred to 20 % as a determinate factor for the city center, why 20 %? I suggest adding justification.

In Paragraph 3 (at the end of the same page) line 2, the author stated that the “previous study could deal…”, which previous study the authors referring to?

Eq. 1 is the basic equation used by the authors for data analysis. I suggest showing an example of the application.

Respond: Firstly, I’m sorry for the confusion caused by this method. In fact, the Natural Breaks (Jenks) mentioned here is a method for classifying layers in GIS. The advantage of this classification method is that the similar values can be grouped in the most appropriate way and the differences between classes can be maximized. The elements will be divided into classes whose boundaries will be set at locations where the difference of data values is relatively larger. Secondly, in terms of the second question, the density of POI on the density surface generated by POI is high in the center of the city and low in the edge of the city, as a result, there would be a difference in POI density surface between the urban center and the urban edge. When the difference exceeds 20%, the difference is irreversible. As for the reason of irreversibility, we have indicated it in the paper. Thirdly, for the third question that you raised, as for the third question you raised, we mentioned the previous research in the literature review, some problems existing in the use of spatial panel model and least squares, and we would like to quote it again here. As for the last question, we have added the theoretical model of GWR in this paper.

5、In the section of the “research results,” paragraph 4, line 2, the author referred to the relative dimension of the urban center as vertical distance. Why vertical distance and not horizontal? Here the author talking about the physical dimension and not a time-space perspective.

Respond: In response to your question, we define the relative size of the city center as the vertical distance. The reason for this is that the difference between the vertical distance of the two cities is more obvious than the difference between the horizontal distance. At the same time, since this chapter mainly discusses the differences between the urban centers of the two cities in terms of morphological dimensions, therefore, this discussion is developed more from the physical point of view, rather than from the perspective of space-time.

6、On the next page of the same section, the author refers to the phrase: “the new normal of china’s economy.” I suggest writing a statement explaining what the “new normal of china’s economy is”.

Respond: The definition of the new normal of China’s economy has been declared and explained in the paper.

 

Reviewer #2:

1、The urban spatial structure is affected by many factors. It is not sufficient to choose the economic development level as the influencing factor. And the selected typical cities Guiyang and Kunming are not representative.

Respond: Indeed, there are many factors that can affect the structure of urban space, including economy, transportation, topography, climate, hydrology and even history and culture. Therefore, it is obviously not comprehensive to analyze the impact of economy, a single factor, on the discussion of urban spatial structure. However, the conclusion of this paper is not that urban economic level plays a decisive or dominant role in urban spatial structure. In this paper, it is concluded that there is a certain two-way relationship between the level of urban economic development and the urban spatial structure, and the form of urban spatial structure will affect the development of urban economic level to some extent. There is no doubt that we are very grateful for your comments. For this reason, we decided to consider more influencing factors in the following study, including influencing factors we mentioned earlier. In addition, we will further judge the influence of different factors on urban spatial structure in the following research. We sincerely hope that you can continue to be our reviewer in our following studies.

When it comes to the study on urban spatial structure and urban economic development level, Mega cities or urban agglomerations like Beijing, Shanghai and Guangzhou may be occurred to many people, the reason for this phenomenon is that the urban spatial structure of these mega cities or urban agglomerations is relatively perfect, and the urban economic level is relatively prominent. It is also the reason that few researchers pay attention to cities in western China. The comparative analysis of case studies on western China that conducted by us can make up for the lack of such research in western China to a certain extent. Secondly, as for the reasons for choosing Kunming and Guiyang as typical representatives are: Since the implementation of the “One Belt and One Road” policy, the urban space institutions and urban economy in Kunming and Guiyang have undergone major changes in just a few years. In addition, the leaders and planners of the two cities all believe that they have benn developing pretty good, the double discussion of the urban spatial structure and urban economic level could help us prove this practical problem.

2、For the urban economic development level and urban spatial structure, the three-year time dimension is too short to reflect its real change. 

Respond: Indeed, three years may be a little short for the study of urban spatial structure and urban economic development level, which is indeed a possible deficiency of this study, so we described it again in the part of discussion. If there are decades of observation time, there is no doubt that great changes could be indeedly seen on the basis of using remote sensing images and interpreting and comparing GDP. Although the time span of this paper is only three years, it still shows that there is a certain connection between urban spatial structure and urban economic development level, so we think this three-year span is still of certain value. Due to the data and time span, this study only discusses whether there is a connection between the urban spatial structure and urban economic development level. In the next study, we are going to analyze the strength and two-way relationship of this connection.

Finally, this study reacquired the POI data for 2016-2018. If you are interested, we can share the data through email and explore the use of it under the urban spatial structure together (ydxh@mail.ynu.edu.cn).

3、Amap is an online map application. Its data acquisition methods and data analysis methods are not described in sufficient detail. Thus, it is impossible for readers to repeat the research according to the article.

Respond: We have made a little supplementary description of the data acquisition method. Although these supplementary descriptions are not enough for the acquisition and analysis of Amap online map data, the acquisition of data is still the most difficult in the field of urban big data research, which involves certain computer knowledge, including machine learning. Therefore, if the detailed description needs to be carried out, a large amount of space will be occupied in this paper, which would inevitably change the focus of the article. In order for subsequent readers to repeat this study, we have uploaded the POI data involved in this study (DOI 10.17605 / OSF.IO / SD59B), which is convenient for readers to download and repeat this experiment, or even improve it.

4、There are many mistakes and omissions in the article, such as the population "499.02 million people".

Respond: Sorry for the series of errors that appeared in the paper, including grammar, structure, etc. we have reviewed the whole paper again, checked all the sentences in the paper, corrected all the errors that are similar to those you found. You can check the details in the paper.

5、The illustrations in the article are not standardized, and the font size is not uniform. Some Chinese characters appear in some parts of the illustrations.

Respond: We are very sorry for the irregularities caused by the careless processing of the pictures, so we spent a lot of time redrawing and typesetting all the illustrations in the paper to make them clearer and more accurate.

 

Reviewer #3:

1、On Fig. 1, it would be better to connect Guiyang and Kunming from China’s map to enlarged individual map of city by ‘arrow’

Respond: We have redrawn Figure 1, and indicated Kunming and Guiyang on the map of China with arrows, which can be viewed in Figure 1 of the paper.

2、The data are Kunming and Guiyang's POI data of 2016 and GDP grid data of 2016-2018: Can you provide citation or website associated with this sentence?

Respond: I have made a brief introduction of the data source website and simple processing methods in the paper. In order to facilitate subsequent scholars to repeat my experiment process, I also uploaded the data to DOI 10.17605 / OSF.IO / SD59B, hoping to make further communication.

 

Reviewer #4: 

1、The multicollinearity of multivariate regression analysis are not addressed.

Respond: The relationship between urban spatial structure and economy is very complicated. As you said, economic development depends on geographic location, population, industry, etc. Urban spatial structure is also affected by many factors such as economic development, transportation, topography, climate, hydrology and even history and culture. So there is no doubt that multicollinearity does occur when multiple factors are put together to judge the correlation between them. However, this study only discusses whether there is correlation between urban spatial structure and urban economic development level, so there may not be obvious multicollinearity problem of multivariate regression analysis. However, the question you mentioned will be taken into consideration in the future multivariate regression analysis of multiple factors between urban spatial structure and urban economic development level. In addition, we have added a description of the deficiencies of this part in the part of discussion.

2、The results is not clear, especially those in Figure 4. Which facror does it represent for?

Respond: We have redescribed the research results involved in the full text, especially the mentioned FIG 4 (which has been modified to FIG 5), the analysis is mainly carried out from the distribution of values in the figure, the agglomeration characteristics of high values, etc. The content of the analysis includes what the content of the research results represents, what value it has, etc. For specific modifications, please see the modifications in the paper.

---

## [Decision Letter · Decision Letter 1]

24 Jun 2020

Research on the Relationship between Urban Economic Development Level and Urban Spatial Structure—A Case Study of Two Chinese Cities

PONE-D-20-08232R1

Dear Dr. He,

We’re pleased to inform you that your manuscript has been judged scientifically suitable for publication and will be formally accepted for publication once it meets all outstanding technical requirements.

Kind regards,

Bing Xue, Ph.D.

Academic Editor

PLOS ONE

Additional Editor Comments (optional):

Reviewers' comments:

Reviewer's Responses to Questions

**Comments to the Author**

1. If the authors have adequately addressed your comments raised in a previous round of review and you feel that this manuscript is now acceptable for publication, you may indicate that here to bypass the “Comments to the Author” section, enter your conflict of interest statement in the “Confidential to Editor” section, and submit your "Accept" recommendation.

Reviewer #1: All comments have been addressed

Reviewer #3: All comments have been addressed

2. Is the manuscript technically sound, and do the data support the conclusions?

Reviewer #1: Yes

Reviewer #3: Yes

3. Has the statistical analysis been performed appropriately and rigorously? 

Reviewer #1: Yes

Reviewer #3: N/A

4. Have the authors made all data underlying the findings in their manuscript fully available?

Reviewer #1: Yes

Reviewer #3: Yes

5. Is the manuscript presented in an intelligible fashion and written in standard English?

Reviewer #1: Yes

Reviewer #3: Yes

6. Review Comments to the Author

Reviewer #1: The manuscript has been improved substantially. The authors' response to my comments and the other comments addressed by the other reviewers is well done. Best Luck.

Reviewer #3: (No Response)

7. PLOS authors have the option to publish the peer review history of their article (what does this mean?). If published, this will include your full peer review and any attached files.

Reviewer #1: Yes: M Salim Ferwati

Reviewer #3: No

---

## [Editor Report · Acceptance letter]

30 Jun 2020

PONE-D-20-08232R1 

Research on the Relationship between Urban Economic Development Level and Urban Spatial Structure—A Case Study of Two Chinese Cities 

Dear Dr. He:

I'm pleased to inform you that your manuscript has been deemed suitable for publication in PLOS ONE. Congratulations! Your manuscript is now with our production department. 

Kind regards, 

on behalf of

Professor Bing Xue 

Academic Editor

PLOS ONE